# Group Intervention Program to Facilitate Post-Traumatic Growth and Reduce Stigma in HIV

**DOI:** 10.3390/healthcare12090900

**Published:** 2024-04-26

**Authors:** Nuno Tomaz Santos, Catarina Ramos, Margarida Ferreira de Almeida, Isabel Leal

**Affiliations:** 1ISPA—University Institute, 1149-041 Lisbon, Portugal; 2CiiEM—Egas Moniz Center for Interdisciplinary Research, Egas Moniz School of Health & Science, Monte da Caparica, 2829-511 Almada, Portugal; 3WJCR—William James Center for Research, ISPA—University Institute, 1149-041 Lisbon, Portugal; margarida.almeida@ispa.pt (M.F.d.A.); ileal@ispa.pt (I.L.)

**Keywords:** post-traumatic growth, stigma, HIV, psychotherapeutic group intervention, psychotherapeutic group program

## Abstract

**Background:** Research on post-traumatic growth (PTG) and HIV is scarce and the relationship between PTG and stigma is controversial. Group psychotherapeutic interventions to facilitate PTG in clinical samples are effective but none exist to simultaneously decrease stigma in the HIV population. The main objective was to evaluate the effectiveness of an intervention in increasing PTG and decreasing stigma in HIV, as well as to explore relationships between the variables. **Methods:** Quasi-experimental design with a sample of 42 HIV-positive adults (*M* = 46.26, *SD* = 11.90). The experimental group (EG) was subjected to a 9-week group intervention. Instruments: CBI, PTGI-X, PSS-10, HIV stigma, emotional expression, HIV stress indicators, HIV literacy, and skills. Multiple linear regression analysis was performed to assess the relationship between the variables. **Results:** There was an increase in PTG and a significant decrease in stigma in all domains and subscales in the EG. Compared to the control group, stigma (*t*_(42)_ = −3.040, *p* = 0.004) and negative self-image (*W* = −2.937, *p* = 0.003) were significant, showing the efficacy of the intervention. **Discussion:** The intervention demonstrated success in facilitating PTG, attesting that in order to increase PTG, personal strength, and spiritual change, it is necessary to reduce stigma and negative self-image. The research provides more information on group interventions for PTG in HIV, relationships between variables, and population-specific knowledge for professionals.

## 1. Background

Even though living HIV+ is no longer deadly, recent research shows that the experience of living with it can still be felt as a traumatic experience [1,2,3,4,5]. Living with HIV has been demonstrated to have several negative psychological effects (e.g., stress and stigma), which can also lead to, on the other hand, post-traumatic growth (PTG, [6]), also known as the perception of positive changes in various areas of the individual’s life as a response to the individual’s confrontation with a potentially traumatic event (e.g., diagnosis and/or experience of HIV). PTG can occur in five domains: appreciation for life, new possibilities, personal strength, spiritual change, and interpersonal relationships [7,8]. Stigma involves experiences of stereotypes, prejudices, and discrimination due to HIV infection, more specifically the internalization of negative feelings and beliefs due to these experiences of discrimination (internalized stigma; [9]). 

The study of the relationship between HIV and PTG has been neglected. There is a possibility that PTG in HIV is mediated by HIV-related stigma, even though the existing evidence has shown inconsistent results. Some studies have shown a direct relationship between HIV-related stigma and PTG [10] or an indirect relationship mediated by social support [11], while one study in particular showed no relationship between stigma and PTG [12].

There are group interventions with the primary objective of facilitating PTG [13]. However, there are no psychotherapeutic group interventions that not only aim to facilitate PTG but also reduce the stigma in people living with an HIV+ diagnosis. Research about group interventions can be controversial in this area. Some recommend the development of new research protocols [13] while others see no added value in this [8]. There is also the fact that the existent validated protocols have not been tested in other populations, with different contexts and particularities (e.g., [14,15]). 

The model of PTG [8] describes the direct and indirect role that several variables can have for the development of PTG to occur in an individual or group intervention In addition to these factors, a psychotherapeutic intervention with the primary goal of facilitating PTG should consider the stages defined for psychotherapeutic intervention based on the PTG model. One example of this is the protocol developed by Ramos et al. [14,15] of a group intervention that included five stages in a particular order (i.e., (i) psychoeducation about the traumatic experience according to the person’s needs, (ii) understanding and training emotional regulation, (iii) cognitive restructuring, communication, and emotional disclosure, (iv) integration of the traumatic experience in the narrative of life, and (v) reconstruction of personal goals and find a new meaning in life) as recommended by Tedeschi et al. [8], with the purpose of promoting individual PTG in women with non-metastatic breast cancer. However, considering that this intervention protocol [14,15] was validated for a specific population and context, it is unknown whether it could also be effective when adapted for other clinical populations, such as people with HIV+.

The main goal of this study is therefore to analyze whether the protocol by Ramos et al. [14,15], when adapted to the HIV population, can also facilitate PTG. To evaluate the effectiveness of the group intervention, whether people with HIV who participate in the experimental group (EG) show more PTG and less stigma than the control group after the intervention will be tested. The research question for this study is therefore the following: what are the effects of a psychotherapeutic group intervention for the facilitation of PTG on the increase in PTG and the reduction in stigma in people living with HIV?

Another purpose of this study is to understand whether there is a relationship between HIV-stigma and PTG. It is expected that this group intervention will increase PTG and decrease stigma and that there will be a relationship between these two primary variables. 

## 2. Methods

### 2.1. Research Design

The method is quantitative, quasi-experimental, and descriptive-observational with pre- and post-intervention measures as well as two groups of HIV+ adults (control group [CG] and an experimental group [EG]). The EG was subjected to a 9-session group psychotherapeutic intervention to facilitate PTG and reduce stigma. Figure 1 illustrates the diagram of the experimental design. The primary variables were PTG and stigma, by total scale and subscale scores. The secondary variables were challenge to core beliefs, perceived stress, HIV-related stress, perceived HIV health literacy and skills, and emotional disclosure.

### 2.2. Participants

Sampling was non-probabilistic by convenience with a snowball effect. The inclusion criteria were (i) over 18 years of age, (ii) diagnosis ≥ 6 months, (iii) no report of mental or physical illness incapacitating them from accessing and completing the questionnaires; (iv) being able to read, write, and comprehend Portuguese; and (v) no diagnosis of schizophrenia or type I bipolar disorder. A total of 64 participants were recruited at the beginning of the study. A total of 42 participants took part in T2, 21 in CG, and 21 in EG. The average age was 46.26 years (*SD* = 11.90, Min = 20, Max = 68). In the CG (*n* = 21), the average age was 46.52 (*SD* = 13.48, Min = 20, Max = 68), while in the EG (*n* = 21) it was 46 years (*SD* = 10.41, Min = 31, Max = 60). The remaining socio-demographic features are displayed in Table 1.

The average number of years with an HIV diagnosis was 15.57 (*SD* = 9.33, Min = 1, Max = 32) in the CG and 16.10 years (*SD* = 16.10, Min = 4, Max = 36) in the EG. Most of the participants in the CG were diagnosed with HIV when they “presented symptoms and were offered a test” (*n* = 8, 38.1%) and in the EG when they were “asymptomatic, and had a routine rapid test” (*n* = 8, 38.1%). The start of antiretroviral therapy occurred mostly in the CG “Between 1 and 6 months after diagnosis” (*n* = 6, 28.6%) and in the EG between “Between 1 and 30 days after diagnosis” or “More than 2 years after diagnosis” (both *n* = 6, 28.6%). The main mode of transmission was unprotected sexual intercourse in the GG (*n* = 21, 100%) and in the EG (*n* = 19, 90.5%). The majority took antiretroviral medication daily. 

It should be highlighted that there were no statistically significant differences between the groups (EG vs. CG) with regard to the socio-demographic and clinical characteristics.

### 2.3. Procedures

The study took place from September 2022 to May 2023 at the premises of associations that support people living with HIV and AIDS in Lisbon, Portugal. Participants were referred by community-based organizations and contacted by the researcher directly (i.e., in person, by telephone, or by email).

Participants were informed of (i) the researcher’s identity and affiliation to the university and partner organization, (ii) the reason for the contact, (iii) what was the purpose and method of the study, and (iv) the added value. They were then invited to sign an informed consent form and voluntarily take part in the study. It was also explained that the confidentiality and anonymization of the data would be guaranteed during the study. The assignment of participants to groups (i.e., CG or EG) was not randomized and participants who expressed an interest in participating in the psychotherapy group were assigned to EG. 

The second evaluation (T2) of the EG was carried out at the end of the last intervention session. The intervention program replicated in this study was authorized by the National Data Protection Commission and was previously registered (ISRCTN02221709). The order in which the assessment instruments were presented was changed from T1 to T2. The data collected were coded (SGIC type) to ensure confidentiality and anonymization.

### 2.4. Intervention

The intervention took place between 15 and 30 days after T1, with 9 weekly sessions (each with a specific theme) and each session taking between 90 and 120 min. The intervention was based on the intervention protocol developed by Ramos et al. [14]. It was necessary to adapt it to the HIV population, which is why an additional session was created specifically to deepen self-awareness about the body with HIV. 

The general aim of the intervention was to increase PTG and to reduce stigma. Each session had specific objectives based on the PTG model [8]. The first session facilitated the development of individual understanding about the emergence of negative emotional responses in reaction to the HIV diagnosis, complemented by some psychoeducation according to the needs of each participant (and above all with a view to promoting HIV health literacy). In the second session, the main objective was to promote disclosure and emotional communication. The third session encouraged the learning of emotional self-regulation techniques. The fourth session facilitated the sharing of HIV-related fears and concerns while continuing to reinforce learning about emotional regulation techniques. In the fifth and sixth sessions, the goal was to identify and understand the gains and losses following the HIV diagnosis and to integrate these gains and losses coherently into each participant’s individual narrative. The seventh and eighth sessions facilitated the development of new values and priorities in life and also included a reflection on the body in HIV. In the ninth session, there was a redefinition of life goals.

### 2.5. Measures

PTG. The post-traumatic growth inventory-expanded tool ([PTGI-X]; [16]) was used. The PTGI-X has not yet been validated for the Portuguese population but has been translated and adapted in this research. A confirmatory factor analysis was carried out, which confirmed the same factor structure for the present sample, as well as good psychometric properties. The PTGI-X has 25 items and five dimensions (i.e., the strength of the person, new possibilities, relationship with others, appreciation of life, and spiritual change) with a six-point Likert scale. The total score ranges from 0–125. The PTGI-X has good psychometric properties, with Cronbach’s alphas excellent properties for the US sample (0.97; [16]). In our study, the Cronbach alpha was 0.95 for the total scale and between 0.73 and 0.87 for the subscales;Stigma. We used the Stigma Scale for People Living with HIV–Reduced Version [17], with 12 items and four dimensions. It assesses levels of personalized stigma, concerns about self-disclosure, concerns about public attitudes, and negative self-image. It is a 4-point Likert type (i.e., “1 = Strongly Disagree” to “4 = Strongly Agree”) and the score ranges from 12–48. The scale has good psychometric properties in the original version and in the Portuguese version (0.70). In this study, the concerns about attitudes in the public subscale had a Cronbach’s alpha < 0.50 at T1 (0.37) and T2 (0.21), so it was removed. An exploratory factor analysis and a confirmatory factor analysis were carried out, which confirmed the same factor structure for this sample, as well as good psychometric properties: 0.86 (Stigma), 0.82 (Personalized Stigma), 0.78 (Disclosure Concerns), and 0.84 (negative self-image);Core Beliefs. The Portuguese version of the Central Beliefs Inventory [18] was used. The CCI has a single factor with nine items and a six-point Likert scale (i.e., “0 = Not at all” to “5 = Quite a lot”). The score ranges from 0–45. The inventory has good psychometric properties, with a Cronbach’s alpha of 0.85. In this study, the Cronbach’s alpha was 0.83;Perceived HIV Stress. In order to assess the level of perceived stress that the participant faced and was facing as a result of living with HIV, two questions were created: “What level of stress did you feel after being diagnosed with HIV?” and “What level of stress do you attribute to living with HIV at the moment?”. The questions had a seven-point Likert scale (i.e., “0 = Not at all stressful” to “6 = Extremely stressful”);Perceived stress. The Portuguese version of the Perceived Stress Scale ([PSS-10] [19]), a 10-item scale that assesses symptoms of stress, with a four-point Likert scale (i.e., “0 = Never” to “4 = Very Often”) was used. Scores ≥ the 80th percentile (cut-off value) indicate pathologies and/or indications of psychological distress (specifically excessive or burnout with psychosomatic risk when scoring in the ≥90th percentile). The scale revealed good psychometric properties, Cronbach’s alpha 0.87 [19]. In the study, the Cronbach’s alpha was 0.79;Perception of HIV literacy. To assess self-perceived HIV-related health literacy, the participants were asked to assess their capability at that moment to (a) access medical or clinical information about HIV/AIDS, (b) understand HIV/AIDS medical information and its meaning, (c) evaluate medical information about HIV/AIDS, and (d) make informed decisions on medical matters about HIV/AIDS (e.g., “What ability do I recognize in accessing medical or clinical information about HIV/AIDS right now?”, “What ability do I recognize in understanding medical information and its meaning about HIV/AIDS right now?”). The answers were based on a seven-point Likert scale, from “0 = Not Able” to “6 = Extremely Able”;Perception of abilities. To assess self-perception of cognitive, emotional, social, and physical abilities, participants were asked to indicate the following on a seven-point Likert scale, from “0 = No ability” to “6 = Extreme ability”: (a) how they think or reflect on situations, (b) how they manage their emotions, (c) how they relate to others, and (d) how they perceive themselves physically.Emotional disclosure. The distress disclosure index (DDI; [20]) and the opener scale (OS; [21]) were applied exclusively to the EG. The DDI has 12 items, with a five-point Likert-type scale, with scores ranging from 12–60, where a higher score indicates a greater tendency for emotional disclosure. The DDI had a Cronbach’s alpha of 0.93 in the original version and a Cronbach’s alpha of 0.85 in this study. The OS, which assesses the ability to disclose their emotions to a significant person, has five items with a five-point Likert-type scale. The OS has a minimum score of 5 and a maximum score of 25; a higher score indicates a greater tendency for emotional disclosure to a significant person, Cronbach’s alpha in the original version is 0.83 and, in the present study, it was 0.89.

### 2.6. Data Analysis

The data were analyzed in IBM SPSS Statistics and AMOS, both with v29. Data on sociodemographic and clinical characteristics were first analyzed using descriptive statistics (frequencies, means, standard deviation, asymmetry, and kurtosis). The Cronbach’s alphas of the psychological variables were then calculated, considering 0.60 to be acceptable [22]. For statistical comparison between groups and between T1 and T2, parametric and non-parametric tests were applied (i.e., *t*-student and Wilcoxon). The level of significance was considered as α ≤ 0.05 and levels of significance between 0.05 ≤ α ≤ 0.10 were considered as marginally significant [23]. Pearson’s bivariate correlations were carried out, also analyzing the direction and magnitude. In an attempt to find explanatory models for the main variables under study, the set of statistical techniques of multiple linear regression analysis (MLRA) was carried out (with stepwise observation).

## 3. Results

### 3.1. Group Intervention Effects

There were significant differences in the EG from T1 to T2: the increase in PTG (*t*_(20)_ = −12.14, *p* ≤ 0.001), in its five domains (appreciation of life *W* = −3.89, *p* ≤ 0.001; new possibilities *W* = −4.02, *p* ≤ 0.001; personal strength *W* = −3.97, *p* ≤ 0.001; spiritual change *W* = −3.94, *p* ≤ 0.001; relating to others *W* = −3.92, *p* ≤ 0.001); reduced stigma (*t*_(20)_ = 3.396, *p* ≤ 0.001); and respective sub-scales (personalized stigma *W* = −4.022, *p* ≤ 0.001; disclosure concerns *W* = −4.022, *p* ≤ 0.001; negative self-image *W* = −4.025, *p* ≤ 0.001). The EG had median scores for emotional expression at T1 (*M* = 33.53, *SD* = 10.37) with the DDI (i.e., minimum 12 and maximum 60), as well as showing a tendency toward weak expression of emotions or discussion of thoughts with significant people on the opener scale (i.e., minimum 5 and maximum 25) (*M* = 13.68, *SD* = 6.25). The comparison between groups found that the EG had lower scores and significant differences in stigma (*t*_(40)_ = −3.040, *p* = 0.004) and negative self-image (*W* = −2.937, *p* = 0.003). There was also a statistical trend toward significant differences in the increase in PTG in EG (*t*_(40)_ = 1.659, *p* = 0.052), personal strength (*W* = −1.685, *p* = 0.092), and spiritual change (*W* = −1.790, *p* = 0.073), as well as in a decrease in personalized stigma (*W* = −1.936, *p* = 0.053) and disclosure concerns (*W* = −1.937, *p* = 0.053) (Table 2).

### 3.2. Contributions of PTG and Stigma

Appreciation of life T1 scores (β = 0.531, *t* = 5.235, *p* < 0.001), stigma T2 (β = 0.510, *t* = 5.200, *p* < 0.001), and the perception of cognitive capabilities T1 (β = 0.410, *t* = 3.872, *p* = 0.002) and T2 (β = −0.435, *t* = −3.873, *p* = 0.002) were predictors of appreciation of life T2 (*F*(1; 21) = 20.739, *p* < 0.001, *r*^2^*_adj_* = 0.846) in the EG (*n* = 21) (Table 3). Spiritual change T2 was explained by T1 (β = 0.509, *t* = 4.473, *p* < 0.001), stigma T1 (β = 0.503, *t* = 4.497, *p* < 0.001), and perception of physical capacities T2 (β = 0.348, *t* = 3.226, *p* = 0.006), with an explanatory variance in the model of around 81% (*F*(2; 20) = 26.044, *p* < 0.001, *r*^2^*_adj_* = 0.807) (Table 3). Despite the significance of the findings, the results should be interpreted with caution due to the small sample size.

## 4. Discussion

The intervention demonstrated success in facilitating PTG, particularly personal strength and spiritual change, as well as in reducing stigma and negative self-image. The results regarding the effectiveness of the intervention in relation to the main objective of this study showed that the protocol [14] adapted to the HIV population facilitates PTG (although the results were marginally significant) and reduces HIV-related stigma when comparing the intervention and control groups. Participation in the group also promoted a reduction in negative self-image. The results can be justified by the techniques used in the protocol, which were developed with the purpose of facilitating a reduction in the levels of stigma and the development of PTG. 

Psychoeducation can increase knowledge about HIV, which can enable the deconstruction of erroneous thoughts and thus improve psychological adjustment in response to stigma. In parallel, psychoeducation can facilitate self-understanding of the physiological and emotional responses that have arisen in the participant’s experience in response to the trauma (i.e., whether in relation to receiving a diagnosis or living chronically with the disease), as well as promote the re-evaluation of core beliefs underlying self-stigma (e.g., in terms of fears and concerns about disclosing HIV-positive status, sexuality, body self-perception, and the infection/disease [8,24]), leading to the development of PTG. Additionally, when psychoeducation is provided in accordance with the participant’s needs, this also allows for an increase in personal competence, which can lead to a feeling of self-confidence (i.e., personal strength), as well as an increase in the desire for meaning from the experience (i.e., spiritual change).

Another technique used in the intervention that can have an influence on the decrease in stigma and the increase in PTG is emotional disclosure in groups. Specifically, disclosure of emotions and feelings related to HIV trauma experience can reduce the negative perception of the disease, facilitating the development of self-acceptance and self-confidence, decreasing symptoms of poorer psychological adjustment (e.g., isolation, guilt, and depression) and thus impacting the core beliefs underlying self-stigma (namely reducing the fear of disclosure concern [25]). The emotional disclosure within a psychotherapeutic group, more particularly the one that occurs through self-disclosure, with the positive effect of acceptance in the group, can also moderate individual adjustment processes of self-acceptance in the incorporation of the HIV/AIDS identity into the self (i.e., including diagnosis, postdiagnosis turning point, immersion, post immersion turning point, and integration [8,26]). Emotional disclosure and the feeling of acceptance in a group can reduce a subject’s feeling of need to protect oneself from stigma [26,27], which can also allow dimensions of core beliefs underlying self-stigma to be actualized through the exchange of experiences [25]. Similarly, emotional disclosure can also lead to the self-exploration of a new identity in the different dimensions of a person’s life (i.e., personal, professional, social, and sexual [28]), which can have some impact on the development of PTG, especially in situations where there has been a loss of trust with intimate partners [29]. In addition, group support or peer support can also influence levels of stigma (i.e., group support tends to decrease stigma as a result of increased social isolation caused by a higher level of stigma [27,28]) and PTG. In a way, social isolation can become a person’s strategy to protect themselves against the aggression brought by stigma and discrimination. While positive group or peer support can influence cognitive–emotional strategies, coping management can redirect toward more deliberate ruminative thinking and the re-evaluation of goals and facilitate the development of scheme change as well as a revision of the trauma narrative for the development of PTG) [8,11]. It seems therefore important to emphasize that the emotional disclosure of EG in T1 showed low levels (i.e., DDI and OS showed low and weak scores), which may have somewhat interfered with better PTG and stigma results at T2, for the reasons we have already explained. Other techniques used in the intervention, such as the exercises of emotional regulation, coping, diaphragmatic breathing, and mindfulness, may also have contributed to the process of reducing stigma and increasing PTG, promoting a better adjustment to the chronic condition [30,31]. These techniques particularly facilitate the reduction in stress levels and the emergence of more deliberate rumination, which leads to PTG and/or self-acceptance of vulnerabilities (personal strength) and/or the search for meaning (spiritual change) in the chronic experience [14,32,33,34]. Expressive writing, the identification of objectives, the construction of an action plan, and the search for meaning and purpose are also techniques used in the remaining sessions of the intervention, which reinforce the decrease in stigma and increase in PTG [8,35,36]. 

Two other relevant aspects to highlight when discussing the results of the intervention are that (i) there was a significant increase in PTG in both groups from T1 to T2, compared to other studies with the same population [10,11,12,37], and (ii) there was an increase in stigma and core beliefs in the CG at T2 while in the EG it decreased (note that in the case of core beliefs, there was a marginally significant difference between groups). 

The difference observed in the increase in PTG in this study compared to others with the same clinical population can be explained by the influence of cultural aspects [16], both in national terms (i.e., it is unknown which HIV/AIDS-related PTG values in the general population have an impact on social identity) and in more specific terms [8] (i.e., there are multiple cultures within social cultures, with different values and beliefs of their own, which influence PTG differently). One other possible explanation for the increase in PTG in both groups is the expert companion factor that was facilitated in the interview at T1 and with the EG during the group intervention [8,38]. Specifically, active and empathetic listening, maximum respect for the traumatic experience, and unconditional acceptance of the participant’s process can facilitate processes of self-acceptance of vulnerabilities (personal strength) that influence the development of PTG [8]. 

The marginally significant difference in the values of PTG, personal strength, and spiritual change in the EG in T2 can be explained by the fact that the protocol (13–14) used as a reference in this study and applied to the EG was validated to only facilitate PTG in a population of women with non-metastatic cancer, a clinical condition that, in a way, can be thought of as limited to a temporal situation at a given time and not as a chronic disease such as HIV. It should be noted that the literature has shown that the development of PTG in people living with HIV can have different trajectories [39]. Thus, as some of the literature has shown, in a process of psychological adjustment, PTG can be understood from several perspectives (i.e., distressed, illusory, or constructive PTG) and trajectories [8,40]. Another explanation could be that the challenge to core beliefs in the EG may have already occurred previously, given the fact that HIV is a chronic experience and considering the average number of years the participants have been diagnosed with HIV [24,26]. 

Another perspective for the increase in stigma at CG in T2 is that stigma might be a broader concept than just HIV-related and that it may intersect with other types of stigma and other factors such as minority stress [41,42,43]. With regard to the objective of understanding possible relationships between stigma and PTG, the results are in line with the literature, which indicates that there is a direct explanatory relationship between the two variables (e.g., [10,44]). 

One possible explanation of the appreciation of life predictive model is that the way the subject thinks and reflects can be positively influenced and updated by the exchange of experiences in a peer group and in safe and positive relationships, which simultaneously make it possible to reduce feelings of stigma (i.e., guilt, shame, and moral judgment) and can increase acceptance of the vulnerable condition of living with chronic HIV (i.e., the present has greater value because there is greater perception that the outcome of the traumatic experience offered a second chance [8]). If there is, for example, greater knowledge about the effects of antiretroviral therapy, there might be a feeling of a second chance at life in the face of the vulnerability of HIV infection. 

On the other hand, one justification for the relationship in the spiritual change model could also be that self-perception of physical capacity (e.g., at the level of core beliefs about one’s own body image and sexuality and the physical progression of the infection/disease [24]), together with stigma factors, moderates concerns about existence and meaning in life in the long term (e.g., in aging). It should also be noted that in both models of T2, the largest influence came from the variables themselves (i.e., appreciation of life and spiritual change) at T1. In a way, it can be thought that valuing life and the existential meaning of life (e.g., physical) in the different life cycles (e.g., adult life or aging) of a chronic illness (e.g., after diagnosis or throughout the chronic experience), such as HIV, can result in various challenges that influence the positive perception of PTG initially achieved. For example, initially after receiving the diagnosis, greater importance might be attached to the fact that antiretroviral therapy offers a higher life expectancy. However, over the course of the chronic experience, greater importance can be attached to the long-term side effects of antiretroviral therapy. This goes in line with the authors of the PTG model [8], which point to a lack of research and the effect of the type of traumatic experience in the development of PTG. In other words, it is possible that the diagnosis of HIV is not the only traumatic experience related to HIV experience and that other experiences such as adaptation and/or resistance to antiretroviral therapy drugs, situations where low immunity occurs, and/or the aging process can also be traumatic for an individual. The perception of stress associated with each of these traumatic experiences is likely to moderate PTG [8]. In the current study, some participants revealed that they had cumulative traumatic experiences with receiving an HIV diagnosis, trauma from sexual violence and abuse, hospital obstetric violence, and job loss due to discrimination. So multiple traumas can interfere with the development of PTG, as research has also shown, and PTG can result from cumulative trauma [45], leading to lower levels of PTG [46].

## 5. Limitations

When observing the results, some limitations should be taken into account, namely that this study was limited to a quasi-experimental design of two assessments, without blinding assessments and randomization between the two groups. The loss of participants (number of drop-outs from T1 to T2 in EG) is another limitation as it could have had an impact on statistical power. In addition, it is recognized that the initial interviews to select participants for the EG may have biased the development of PTG. It is also acknowledged that there may have been attrition on the part of the participants when completing the questionnaire in T1. Another relevant aspect is the small sample size and convenience sampling, which makes it difficult to generalize the results to the HIV population. A Randomized Controlled Study with a larger sample is suggested in order to clarify and support the results found here. Even though the two explanatory models found a high explanatory variance (i.e., 85% and 81%), future studies could also include and explore other variables (e.g., psychological adjustment, identity, core beliefs, body self-consciousness, physical well-being, sexual self-esteem, and expert companion) as well as sociodemographic and HIV-related variables that might be related to PTG and stigma in the HIV population.

## 6. Clinical Implications 

This study makes a significant contribution to PTG and stigma research, specifically in the area of group intervention and in the HIV population. To our knowledge, this was the first study to explore the possibility of adapting a group intervention protocol to facilitate PTG as a primary goal and reduce stigma in a population with HIV, which points to the versatility of the Ramos et al. [14,15] protocol. It was also the first to report on emotional expression results (i.e., expert companion) with a clinical sample in HIV while contributing to the empirical evidence of the connection between the constructs of PTG and stigma. There is a growing body of evidence that shows how health literacy in HIV is important. This study contributes to increasing the knowledge about the HIV+ clinical condition and population, for any psychologist or healthcare professional who provides care and intervenes with this population, by drawing attention to various constructs, which can ultimately improve clinical interventions in individual and group HIV care.

## 7. Conclusions

The intervention demonstrated success in facilitating PTG, attesting that in order to increase PTG, personal strength, and spiritual change, it is necessary to reduce stigma and negative self-image. The research provides more information on group interventions for PTG in HIV, relationships between variables, and population-specific knowledge for professionals.

## Figures and Tables

**Figure 1 healthcare-12-00900-f001:**
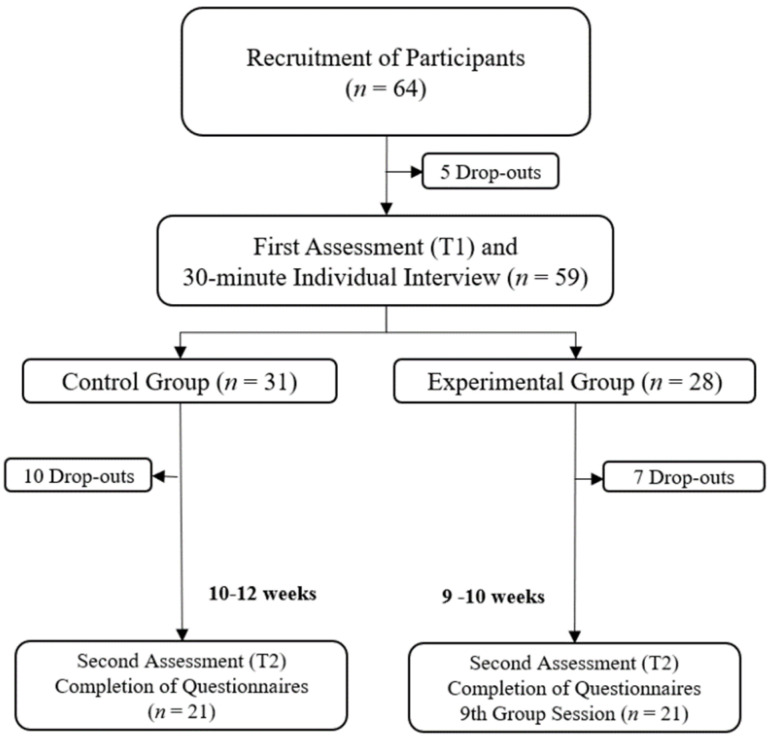
Diagram of the quasi-experimental design.

**Table 1 healthcare-12-00900-t001:** Sociodemographic characteristics in CG (*n* = 21) and EG (*n* = 21) in T1.

Variable	CG (*n* = 21)		EG (*n* = 21)	
	*n*	%	*n*	%
Age, years (*M*, *SD*)				
Nationality	46.52	13.48	46	10.41
Portuguese	16	76.2	17	81
Residence				
Lisbon	16	76.2	14	66.7
Ethnicity				
White/Portuguese and/or Of/European Origin	16	76.2	16	76.2
Sex				
Female	9	42.9	9	42.9
Male	12	57.1	12	57.1
Sexual Orientation				
Heterosexual	10	47.6	10	47.6
Gay	11	52.4	11	52.4
Academic Qualifications				
9th Year	7	33.3	4	19
12th Year	3	14.3	6	28.6
Degree	4	19	7	33.3
Professional Status				
Working	9	42.9	9	42.9
Unemployed	6	28.6	9	42.9
Retired	6	28.6	3	14.3
Marital status				
Single	15	71.4	13	61.9
Love relationship				
Yes	6	28.6	9	42.9
Has children				
Yes	7	33.3	8	38.1
Before HIV Diagnosis	3	14.3	5	23.8
After HIV Diagnosis	4	19	3	14.3

Note: CG = control group; EG = experimental group; *n* = number of occurrences observed in the sample; % = relative frequency 0–100%.

**Table 2 healthcare-12-00900-t002:** Mean, SD, and statistical differences between EG (*n* = 21) and CG (*n* = 21) in T2.

Variable	EG		CG		TS	*p*
	*M*	*SD*	*M*	*SD*		
PTG	88.95	20.44	76.48	27.75	1.659	0.052
Appreciation of Life	11.76	3.32	9.90	4.37	−1.558	0.119
New Possibilities	18.90	4.47	16.24	6.31	−1.427	0.154
Personal strength	16.67	2.80	14.62	4.08	−1.685	0.092
Spiritual Change	20.24	7.04	16.95	7.34	−1.790	0.073
Relating to Others	21.38	9.14	18.76	9.33	−0.856	0.392
Stigma	26.48	6.77	33.48	7.06	−3.040	0.004
Personalized Stigma	1.86	0.86	2.41	.95	−1.936	0.053
Disclosure Concerns	2.63	0.92	3.19	.85	−1.937	0.053
Negative self-image	6.00	2.83	6.05	2.84	−2.937	0.003
Core Beliefs	2.97	1.38	3.30	1.02	−0.879	0.192
Perceived Stress	20.57	7.97	23.67	5.59	−1.457	0.076
Perceived HIV Stress						
After diagnosis	4.71	1.77	5.05	1.56	−0.794	0.427
Today	1.81	1.75	2.52	1.60	−0.181	0.856
HIV health literacy						
Access information	5.05	1.07	4.29	1.38	−1.825	0.068
Understanding information	4.57	1.25	3.71	1.65	−1.792	0.073
Evaluating information	4.19	1.66	3.52	2.09	−0.967	0.334
Making decisions	4.05	1.56	3.57	2.01	−0.720	0.472
Perception of abilities						
Cognitive	4.00	1.05	4.57	1.29	−1.585	0.113
Emotional management	3.52	1.29	3.52	1.44	−0.026	0.979
Social	3.71	1.42	4.00	1.48	−0.863	0.388
Physical	3.76	1.30	3.86	1.20	−0.388	0.698

Notes: T2 = after intervention; EG = experimental group; CG = control group; *M* = mean; *SD* = standard deviation; TS = *t*_(41)_ = *t*-student test statistic and degrees of freedom or W = Test Wilcoxon; *p* = significance.

**Table 3 healthcare-12-00900-t003:** Explanatory models of PTG and stigma in EG (*n* = 21) in T2.

Variables/Model	β	*T*	*p*-Value	*F*	*p*-Value	*r* ^2^ * _adj_ *
Model: Appreciation of Life (DV) (*n* = 21)				20.739	<0.001	0.846
Appreciation of Life T1	0.531	5.235	<0.001			
Stigma T2	0.510	5.200	<0.001			
Cognitive T2	0.410	3.872	0.002			
Cognitive T1	−0.435	−3.873	0.002			
Social T1	0.271	2.190	0.047			
Model: Spiritual Change (DV) (*n* = 21)				26.044	<0.001	0.807
Spiritual Change T1	0.509	4.473	<0.001			
Stigma T1	0.503	4.497	<0.001			
Physical T2	0.348	3.226	0.006			

Note: The variables present in the models were core beliefs, post-traumatic growth and subscales, perceived stress, perceived stress related to HIV after diagnosis and today, stigma and subscales, literacy indicators, and perceived capacity, except in cases where the variable was indicated as DV.

## Data Availability

Data are contained within the article.

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
