# Peer review of "Group Intervention Program to Facilitate Post-Traumatic Growth and Reduce Stigma in HIV"

_healthcare, 2024, doi:10.3390/healthcare12090900_

Round 1

Reviewer 1 Report

Comments and Suggestions for Authors

Thank You very much for the opportunity to read this interesting and unique scientific study. Positive psychology of rehabilitation in the event of acquiring a disability is primarily interested in building mental strength, creating meaning, coping, strengthening immunity, and post-traumatic growth.

Therefore, the key to positive rehabilitation psychology is to strengthen those resources that help cope with the disease, endure the physical and mental stress associated with it, and carry out the most important life tasks despite obstacles and limitations. The task of positive rehabilitation psychology is to guide somatically ill people towards constructive adaptation and post-traumatic growth. The reviewed article fits perfectly into this context.

Importantly, in people with disabilities or somatic illnesses, the simultaneous presence of positive and negative aspects of human functioning can be particularly clearly observed. A person with a serious bodily injury may experience self-efficacy, satisfaction from overcoming difficulties or happiness from close relationships, simultaneously with experiencing pain and a sense of loss, as well as, for example, simply anger at their own powerlessness and limitations. I appreciate the research plan very well illustrated in Figure 1 and the research procedure, which takes into account all important aspects of inference based on post-training or post-therapeutic data.

Pointing out how important post-traumatic development is does not mean that it is a "better" or more important perspective than an approach focused on correcting deficits. It is important to use both approaches and thus create the fullest possible picture of a given person's situation, and thus conduct as comprehensive interventions as possible. The authors of the study reported in the article managed to combine these two perspectives.

The article submitted for review is interesting, and after revision and reassessment by the reviewers, it seems possible to publish it.

Notes to the presented text are as follows:

1. The abstract should correspond to the content of the article to a greater extent. It should contain more specific assumptions, a description of the method of analysis used and the most important conclusions.

2. The theoretical background of the interesting research proposed by the Authors should take into account to a greater extent contemporary research on post-traumatic growth in clinically comparable diseases. The Authors' correct comment about the lack of similar research on HIV does not justify a rather superficial presentation of the research assumptions.

This fragment should also be enriched with more modern theories and research relating to the research problem undertaken in the further part of the study. Despite this reservation, the introduction is coherent, logical and scientifically valuable.

3. The most important objection to the reviewed text concerns the method of analyzing the results of own research.

a) The article does not present research questions or research hypotheses. In fact, therefore, this fact alone should disqualify the text. The analysis of the results de facto does not meet the basic methodological criterion in studies conducted in the nomothetic approach. The Authors should correct this weakness in the text. In general, the research was very well prepared and carried out using a quasi-experimental procedure, so I propose publishing the text after corrections.

b) The T1 and T2 examination time markings are intuitive, but Table 1 lacks appropriate column markings. It is worth supplementing them.

c) The Authors' very bold conclusions based on the model of explaining PTG and stigma raise some reservations. The direction of the analysis is correct, but limitations result from the low number of the study group.

4. The discussion of the results lacks a clear direction in the selection of sources and a clear presentation of the conclusions from the study.

Conclusions:

Posttraumatic growth means that a person is functioning better than before the traumatic experience or extremely challenging situation. The mechanism responsible for this positive change is the reorganization of a person's cognitive structures - the way of thinking about oneself, life and the surrounding world changes.

I recommend that the Authors make the correction indicated in the review and then - after another review - publish an interesting and research-important text. Congratulations on an extremely interesting and socially necessary scientific project. It is worth extending the research in the future or carrying it out taking into account new variables - theoretically important from the perspective of PTG.

In my opinion, the text of the article lacks a specific reference to the decision of the research ethics committee. It should be verified whether such an opinion has been issued, because the group of HIV patients is particularly vulnerable.

Author Response

Good afternoon,

Thank you for reading our article and for your suggestions for improvement. We have responded to your suggestions below:

1) Thank you for your suggestions. We have tried to improve, but we are limited to the number of words in the section. The hypotheses are indicated, i.e.: i) it is indicated “The main objective was to evaluate the effectiveness of an intervention in increasing PTG and decreasing stigma in HIV, as well as to explore relationships between the variables", ii) the methods are also explained “Quasi-experimental design, a sample of 42 HIV-positive adults (M =46.26, SD = 11.90). The Experimental Group (EG) was subjected to a 9-week intervention. Instruments:  CBI, PTGI-X, PSS-10, HIV stigma, emotional expression, HIV stress indicators, HIV literacy and skills and iii) the general conclusions are also mentioned.

2) In fact, research on diseases that are clinically comparable to HIV is lacking. However, since the introduction of antiretroviral therapy (i.e., around 1995), it has been possible to think about and study living with HIV in a way that is comparable to other diseases, especially chronic ones. Only recently has it become clear that the challenge of living with chronic HIV is primarily the early onset of comorbidities and long-term neurocognitive problems, which can vary depending on the clinical and developmental history of the individual. As a disease, this makes it different from other chronic diseases. Unfortunately, this specificity is still being explored, as the introduction of antiretroviral therapies has also led to a number of changes in the types of medications. On the other hand, in the theoretical field of PTG research, aspects such as the change in core beliefs during the experience of a chronic illness, regardless of the illness, are still little studied. Therefore, in this study, we have tried to avoid associations with other illnesses, sticking only to what exists in research with direct links.

3) 

a) Thanks for your comment. The research question was included, as requested. “The research question of this study is as follows; What are the effects of a psychotherapeutic group intervention for the facilitation of PTG on the increase of PTG and the reduction of stigma in people living with HIV?” This research has struggled with research gaps on the specifics of the topics covered. As has been shown, it is possible to present research questions and even specific objectives, but not hypotheses, because of the gaps in the literature. This is why we chose not to detail hypotheses, only objectives. We hope that this will no longer be an issue in future articles.

b) Thank you for your suggestions. In fact, we noticed a flaw in the identification of the first two tables. We have corrected it

c) We agree that the number of participants was indeed a major limitation. We hope that in future articles, this issue can be improved .Accordingly to your suggestion, we have included this sentence in the end of the Results section: “Despite the significance of the findings, the results should be interpreted with caution due to the small sample size.”

4) Thank you for your suggestion. The lack of research into the specifics of the themes studied in this study makes it difficult to draw overly specific conclusions about the research, however, we have improved the discussion by adding the following sentence: in the beginning of the Discussion section “The results regarding the effectiveness of the intervention in relation to the main objective of this study showed that the protocol [13] adapted to the HIV population facilitates PTG, (although the results were marginally significant) and reduces HIV-related stigma when comparing the intervention and control groups. Participation in the group also promoted a reduction in negative self-image."

Thank you for your pertinent comment. Based on your suggestion, we have added a clarifying sentence to the procedures section. “The intervention program replicated in this study was authorized by the National Data Protection Commission and previously registered (ISRCTN02221709)."

Reviewer 2 Report

Comments and Suggestions for Authors

This is important work, seeking to extend the concept of posttraumatic growth and stigma to those with HIV.

The Background section is adequate but is not especially strong. This section would benefit from an explicit definition of posttraumatic growth and stigma. A bit of history on the development of the concept of "posttraumatic growth" would be useful. While the reference list has 39 sources, I noted only a few are recent. There is a good description of Ramos et al.'s protocol.

In the Methods section (2), the first sentence does not have a beginning. Figure 1 is very good in illustrating the research design and the experimental and control group samples. The subheading 2.2 Participants is very good. Table 1 is a nice illustration of participant characteristics. Is there an explanation for why the Control Group presented with symptoms and the Experimental Group did not? Was this purposeful in the study? The Procedures and Measures sections are both very good.

The Measures section (2.5) is especially strong. Each of the 8 inventories/scales is well-described. I'm impressed the authors were able to utilize 8 measures with their sample. It is especially important that they included the self-measure of HIV literacy.

I am not a research methodologist/statistician and I hope another reviewer can be helpful here. I hope attention can be given to the use of the phrase "a statistical tendency to be significant" in line #272.

Line #232 "there" should be "There"

In the Discussion section, it is not a surprise that psychoeducation can be useful. The discussion of the role of groups is especially strong and very meaningful in the context of an HIV diagnosis. Cultural aspects are also very important to include.

Line #288 "in" should be "is"

Not sure why it is important to include a comparison with women with non-metastatic breast cancer.

It is important to address both spiritual change and appreciation of life and both are well-described.

The question becomes: What does postraumatic growth overall look like for those with HIV? What is missing for me is a concise picture of what the participants in the experimental group have gained. A short paragraph at the end of the Discussion section could provide that.

The Limitations and Clinical Implications sections are good. The last line of the article needs to be revised to make it a complete sentence.

Comments on the Quality of English Language

Quality of English language is fine.

Author Response

Good afternoon,

Thank you for reading, for your suggestions and questions.

Thank you for your pertinent comment. As suggested, we have added a brief definition of PTG and stigma in the introduction, taking into account the word limit. For PTG, we added “Posttraumatic Growth (PTG, [7]), also known as the perception of positive changes in various areas of the individual's life in response to the individual's confrontation with the potentially traumatic event (e.g., diagnosis and/or experience of HIV)..” For stigma "involves experiences of stereotypes, prejudices and discrimination due to HIV infection, more specifically the internalization of negative feelings and beliefs due to these experiences of discrimination (internalized stigma; [36])."

As suggested, we included the symptoms of the experimental group in the manuscript.

The literacy questions were constructed by us and are indicators (i.e., individual questions), not a questionnaire or scale. However, we have included two sample questions in the Instruments section: "What ability do I recognize in accessing medical or clinical information about HIV/AIDS right now?", "What ability do I recognize in understanding medical information and its meaning about HIV/AIDS right now?"

Line #232 "there" should be "There": Corrected.

Thank you for your comments. Cultural aspects are very important, however, given the objectives of the research, and the fact that the PTGI-X is still in the process of being validated for Portugal, we think it would be premature to present arguments. Although there may be cultural differences in the dimensions of PTG between populations, the phenomenon itself is universal.

With regard to how the PTG, as an experience, was felt by the EG participants, for the time being, and given that the methodology is quantitative, it seemed inappropriate to detail more information.

Once again, we thank you for your readership and suggestions.

Reviewer 3 Report

Comments and Suggestions for Authors

Thank you for the interesting work. See my comments and suggestions below:

-              Please describe your participant recruitment and group assignment processes

-              Add more details for Table 1, specifically what tests you used to measure any statistical differences between EG and CG. Also, losing your participants affected power and if there was missing at random in the method section. You mentioned in the texts that there were no statistical differences between the groups (LL: 112-113), however, it looks like there were more unemployed and in relationship in EG (42.9 % vs 28.6%).  Lack of statistical power can be caused by a small sample size (e.g.: https://www.ncbi.nlm.nih.gov/pmc/articles/PMC4296634/), please address this issue in your discussion section. 

-              Please describe your intervention in more detail

-              Please add a table with pre-post changes in EG and CG and indicate changes mentioned in the text (it is difficult to read the text without a table) as well comparison EG and CG at T1

-              Please clarify if you added demographic variables to your final model and provide your rationale in any case. 

-              Provide a rationale for including both T1 and T2 measures of the same variables in your models 

-              Please pay more attention to language as some sentences do not look complete. 

Comments on the Quality of English Language

Some sentences need revision as they do not seem complete. 

Author Response

Thank you for your relevant comment. The information about participant recruitment was included in the manuscript as follows: “The assignment of participants to groups (i.e., CG or EG) was not randomized, and participants who expressed an interest in participating in the psychotherapy group were assigned to EG.” In fact, given the limited availability of participants for the psychotherapy group, it was not possible to randomize participants, so we accepted all those who expressed interest in participating in the EG.

Thank you for your suggestions. The identification of the statistical tests was placed, as suggested, at the end of the table in a legend. “t(41) = t-student test statistic and degrees of freedom; W = Test Wilcoxon; p = significance;”

Thank you for your comment. This information was added to the Limitations in the Discussion section. We hope it addresses your concerns.

With regard to "In addition, losing your participants affected the power and whether there was a random lack in the method section": Thank you for your comment. This information was added to the Limitations in the Discussion section. We hope it addresses your concerns.

The issue of statistical power has been reinforced, especially due to the limited size of the sample.

We have described the intervention in more detail. See the text below:

“The general aim of the intervention was to increase PTG and to reduce stigma. Each session had specific objectives based on the PTG model [8]. The first session facilitated the development of individual understanding about the emergence of negative emotional responses in reaction to the HIV diagnosis, complemented by some psychoeducation according to the needs of each participant (and above all with a view to promoting HIV health literacy). In the second session, the main objective was to promote disclosure and emotional communication. The third session encouraged the learning of emotional self-regulation techniques. The fourth session facilitated the sharing of HIV-related fears and concerns, while continuing to reinforce learning about emotional regulation techniques. In the fifth and sixth sessions, the goal was to identify and understand the gains and losses following the HIV diagnosis, and to integrate these gains and losses coherently into each participant's individual narrative. The seventh and eighth sessions facilitated the development of new values and priorities in life, and also included a reflection on the body in HIV. In the ninth session, there was a redefinition of life goals.”

Regarding the question "Please add a table with the pre-post changes in the EG and CG and indicate the changes mentioned in the text (it is difficult to read the text without a table), as well as the comparison of the EG and CG at T1": Thank you for your suggestion to include another table. However, given the limited number of words, we decided to describe the differences between groups at T1 in the text (rather than in a table). This was done to simplify the information and to avoid repeating it. If necessary, we are willing to change this.

Regarding the question “Please clarify if you added demographic variables to your final model and provide your rationale in any case”: Thank you for your comment. Considering the aims of the study, we decided to build explanatory models only with the psychological variables. However, it was added in the discussion section that the inclusion of other sociodemographic and clinical variables in the explanatory model is recommended.

Regarding the statement "Please pay more attention to language as some sentences do not look complete": we appreciate the suggestion, we have re-read the whole article and tried to improve it.

Round 2

Reviewer 3 Report

Comments and Suggestions for Authors

Thank you for considering and responding to my suggestions.

Comments on the Quality of English Language

Please make sure to correct typos in the text.